# The Influence of SARS-CoV-2 Pandemic on TMJ Disorders, OSAS and BMI

**DOI:** 10.3390/ijerph19127154

**Published:** 2022-06-10

**Authors:** Sabina Saccomanno, Stefano Saran, Martina De Luca, Rodolfo Francesco Mastrapasqua, Luca Raffaelli, Luca Levrini

**Affiliations:** 1Department of Health, Life and Environmental Science, University of L’Aquila, Piazza Salvatore Tommasi, 67100 L’Aquila, Italy; 2Department of Human Sciences, Innovation and Territory, School of Dentistry, Postgraduate of Orthodontics, University of Insubria, 21100 Varese, Italy; ssaran@studenti.uninsubria.it (S.S.); luca.levrini@uninsubria.it (L.L.); 3Dental School, Catholic University of the Sacred Heart, 00168 Rome, Italy; martina.deluca04@icatt.it (M.D.L.); luca.raffaelli@unicatt.it (L.R.); 4ENT Department, Rivoli Hospital, ASL TORINO 3, 10098 Torino, Italy; rfmastrapasqua@aslto3.piemonte.it

**Keywords:** SARS-CoV-2 pandemic, body mass index, obstructive sleep apnea syndrome, temporomandibular joint disorders, orofacial pain

## Abstract

The pandemic of the 21st century had a significant influence on the lives of the world population in a negative way. This situation determined a change of lifestyle; it caused the necessity of social isolation for a great number of people. In fact, people tended to avoid crowded environments, social events, to reduce medical checks and sports activities, favoring sedentary life because of fear of the virus. This social attitude brought a high level of stress that worsened many health conditions. This study has the aim of evaluating the possible influence of the pandemic on temporomandibular joint (TMJ) disorders, obstructive sleep apnea syndrome (OSAS) and body mass index (BMI). An anonymized survey, available in two languages (Italian and English), was given to 208 patients from different private dental practices. In this questionnaire, the patients shared experiences about their life during the pandemic. The article highlighted that during this health emergency, there was an increase in body weight in the considered sample. This brought a worsening of OSAS in 65% of patients with a previous diagnosis. Eventually, an increase in TMJ disorders and orofacial pain was reported.

## 1. Introduction

In December 2019, the new coronavirus SARS-CoV-2 developed in Wuhan, China. Due to its high transmissibility, the new virus rapidly spread from China throughout the world, causing a pandemic named COVID-19 [1,2]. The clinical spectrum of this virus infection is wide, from asymptomatic and mild cases to atypical pneumonia, to severe respiratory failure and death. It became quickly apparent that SARS-CoV-2 was able to cause cardiovascular injuries such as myocarditis, pericarditis, arrhythmias, infarction, heart failure or thromboembolic events [3]. The exponential growth of COVID-19 cases challenged the capacity of healthcare systems so that governments were forced to impose a lockdown with restrictions concerning all life aspects [4]. Social distancing was enforced, and sports venues, restaurants and cultural facilities were closed. Lockdown measures have strongly influenced life habits everywhere, with an important impact on eating behaviors and sports practice [5]. Recent studies have shown that daily food intake has increased [4], sleep time increased and screen time increased significantly [6], while sports activities decreased. Moreover, long isolation results in increased stress levels, while stress is a risk factor for different pathologies and unhealthy habits. For example, the relationship between temporomandibular joint (TMJ) disorders and stress is well established [7,8,9]. Furthermore, it is known that stress can be related to alcohol misuse [10,11], which contributes to weight gain [12] and impaired food intake [13]. It was observed that during the COVID-19 pandemic, the prevalence of overweight and obesity has significantly increased [14] and almost 65% of individuals affected by eating disorders experienced worsening symptoms during the confinement. Only 16% reported mild or moderate improvement of symptoms [15]. About 50% of individuals with obesity reported increased weight and increased snacking but reduced physical activity. This can be related to anxiety and depression, which were common in the populations during this historical moment [15].

The scientific literature agrees on obesity as one of the most important risk factors for obstructive sleep apnea syndrome (OSAS), a common sleep disorder with complete or partial airway obstruction, disrupted sleep and excessive daytime sleepiness. Cardiovascular, metabolic and neurocognitive comorbidities can also occur, in both the pediatric and adult populations, when left untreated [16]. Furthermore, obstructive sleep apnea seems to be proportional to the degree of obesity [17,18]. A 2016 study showed that OSAS prevalence varies by body mass index (BMI): 32.4% in morbidly obese patients, 20.3% in severely obese patients, 15.7% in obese patients, 9.2% in those who were overweight and only 5.7% in those with normal weight [19].

Our study aimed to analyze, through a questionnaire, the influence of the pandemic on many aspects of human health. How the increase in stress experienced during this pandemic determined the worsening of many people’s medical problems such as obesity, TMJ disorders, dental conditions and respiratory problems, such as OSAS, was explored.

## 2. Material and Methods

An anonymized survey, accessible in both Italian and English, was filled by 208 patients from different private dental practices. In this questionnaire, the patients answered the questions about their life during the pandemic. The survey reached 208 patients from different nations that were included in the study design. There were no reminders transmitted to patients to help them feel free to answer. It was specified that the purpose of the questionnaire was to find ways for clinicians to improve their skills in treating patients. It was ascertained that each patient provided one answer by controlling the timing and different kind of responses. The patients were asked to complete the questionnaire without any possible compensation or benefit in return. The questionnaire was compiled specifically for this study, and due to the contingency of the COVID-19 pandemic waves, pre-testing was not a possible option. We closed the study in January 2022, at the end of the pandemic wave, and recruited a total of 208 responders between October 2021 and January 2022. All participants signed informed consent and accepted the privacy policy for the protection of personal data before completing the survey. No personal information that can identify the individuals was collected and the data were analyzed in aggregate form only. All responses were anonymized using the Google Form service. The resulting data file, used for data analysis, was free of any identifiers, including email and IP addresses or other electronic identifiers. The study was carried out in accordance with the principles established in the Declaration of Helsinki.

The following information was gathered in the questionnaire:GenderAgeEducationProvenienceEmployment statusWeightHeightSmoking habitSARS-CoV-2 infectionNeed of a dentistReasons to need dental therapiesPresence of TMJ disorders/orofacial pain and its possible correlation with eating habitsPresence and possible worsening of OSASPossible influence of weight/diet on oral healthFollowing a specific dietBreathing problems while sleeping or lying on bedOrthodontic treatmentKind of meds more used

Statistical analysis was carried out using Statistical Package for Social Sciences (IBM SPSS Statistics for Windows, Version 25.0. Armonk, NY, USA: IBM Corp). Binary variables such as gender or questionnaire answers were tested for correlation using the chi-square test while multiple-choice questions were tested using the Spearman correlation coefficient.

## 3. Results

The sample analyzed consists of 208 statistical units: 154 women, 51 men and 3 people refused to specify their gender (Table 1). The mean age of the sample is about 44 years old, with a SD of 13.629. The youngest interviewed is 11 years old, while the oldest one is 77 years old. The median is 46 years old. Most of the sample have a university education (Figure 1). Most of the interviewees are from Italy and from Europe in general (Table 2).

A total of 14.6% of the sample smoke. The minimum observed weight is 30 kg, and the maximum observed weight is 204 kg. The mean is about 83 kg. The minimum and maximum height in the sample are 125 and 195 cm, respectively. The mean and the median are about 167 cm. The average BMI is 29.6917 (Figure 2).

A total of 57.6% of the sample have noticed a change in weight during the COVID-19 pandemic. More specifically, 22.9% have gained weight and 4% have lost weight. A total of 32.8% didn’t specify what the change was. A total of 75.5% reported that they were not following a specific diet. Most of the people in the sample did not get infected with COVID-19; only 54 people reported to having been infected (Table 3). Among those who had COVID-19, the majority said that the infection did not influence their weight, 24.1% said that it increased their weight and 18.5% said that COVID-19 infection decreased their weight (not significant, *p* > 0.05). (Table 4).

A total of 128 people (63.4% of the sample) reported that they needed a dentist during the pandemic (Table 5) and 82.7% of them have gone to see one (Table 6). Most of the subjects in the sample (51.9%) reported that they do not suffer from orofacial/TMJ pain (Table 7). A total of 61.6% of those who suffer from orofacial/TMJ pain do not feel that the TMJ/orofacial pain is correlated with what they eat (Table 8). A total of 51.8% of those who suffer from orofacial/TMJ pain said that their TMJ/orofacial pain increased during the pandemic (Table 9). Most of the people in the sample do not have a diagnosis of OSAS, but among those who do, 65.2% noticed an influence of their weight and diet on their sleep breathing quality (Table 10). The majority of the sample report not having problems with breathing while laying/sleeping in bed (74.5%). The majority (60%) of people who reported having OSAS also reported that weight and diet influence this medical problem (*p* < 0.05). Almost half (40.9%) of the people indicated that their oral health is also influenced by changes in weight and diet, while 40% of people who got COVID-19 think that it negatively influenced their orofacial/TMJ health. A total of 30% had an orthodontic therapy during the pandemic, but 66.7% did not feel that the orthodontic treatment improved their general health condition (not significant, *p* > 0.05). During the pandemic, a wide variety of meds were used; the people in the sample reported an important use of anxiolytic and anti-depressants meds. Finally, even if it is of minority importance, this questionnaire highlighted that during this health emergency, people bought foods, clothes and technological devices, respectively, 65.9%, 14.9% and 11.5%, in a major way.

## 4. Discussion

The lockdown period has significantly influenced people’s lives, causing changes in daily habits and leading, sometimes, to a significant increase in stress-related disorders [20]. In this study, the research focused on the appearance or worsening of OSAS due to weight gain linked to the change in habits and the emotional state of people. Furthermore, the worsening of the temporomandibular joint and muscular disorders was assessed, this pathology is already notoriously related to stress, but it was contextualized in this particular historical moment. Through the answers to the questionnaire, it was highlighted that 22.9% of the interviewed subjects noticed an increase in their body weight, compared to a minimum percentage in which a weight loss occurred (4%). This can be caused by a change in lifestyle; an increase in the sedentary life with an important reduction in sports activities. Body mass index (BMI) is a widely used parameter to obtain a general assessment of body weight. Using a simple mathematical formula height and weight of the subject can be related. The weight of the subject, in kg, must be divided by the square of his/her height, expressed in meters. The result of this formula classifies the subject in an area in which weight can be: underweight, normal, overweight, medium degree obesity, high degree obesity. It should be noted that the average BMI of the interviewed population is 29.7 kg/m^2^. A BMI between 18.5–24.99 kg/m^2^ is considered the standard weight [21]. Therefore, our sample was reportedly overweight (25–29.99 kg/m^2^). In addition, we considered that a recent study found that people with a normal BMI did not alter their consumed food amount as much as people with a BMI > 25 kg/m^2^ [4]. Although most of the individuals interviewed said they did not have a diagnosis of OSAS, 65% of those who did suffer from this disorder noticed an influence of diet and weight change on their symptoms. This confirmed what is already said in the literature: it is very important to suggest a healthy lifestyle and diet to people with a diagnosis of OSAS [22]. In fact, it is confirmed that the Mediterranean diet was able to reduce the apnoea-hypopnoea index (AHI) and general health [22].

While 44.2% of the population interviewed reported suffering from TMJ or orofacial pain, 38.4% related these symptoms to the type of food consumed. Some kinds of food can increase muscle activity and the load on the joints, worsening the common symptoms of TMJ disorders. In general, an unhealthy diet—rich in sugars, salt, saturated fats and highly processed—can influence general inflammatory conditions, according to the recent literature, and when TMJ pain is linked to an inflammatory disease, eating habits can influence this condition. Relevant data is the worsening of these pathological conditions reported by more than half of the surveyed patients during the quarantine period, possibly because stress and boredom during a lockdown have been shown to increase intake of “comfort food,” usually “junk food” with poor nutritional value but known to have inflammatory and weight gaining properties [23]. According to what it is written before, it is very important to advise a sugar-free or a very low intake of refined sugar to patients with chronic conditions such as the ones with TMJ disorders. Moreover, the recent literature highlights how a gluten-free diet can be helpful in people with chronic and refractory pain in masticatory muscles [24]. Furthermore, it is interesting to note that 33.4% of the responders who reported being contaminated with COVID-19 also reported an influence of the infection on their TMJ condition. Thus, it is possible that, in addition to conditions of stress and anxiety, COVID-19 infection can also have a negative influence on temporomandibular joint disorders and orofacial pain. Even if more research is needed, it seems that COVID-19 infection generated persistent symptoms also after healing [25]. It is evident the pandemic caused a high level of socio-economic pressure that could generate anxiety or depression and that explains the increase in the intake of meds that can alleviate those kinds of feelings, but in some cases, these emotions determine a problem called panic buying, which is a behavior that brings people to buy in an exaggerated way [26]. This problem can affect men and women, and according to this questionnaire, during the health emergency, the main items bought were food, technological devices and clothes.

## 5. Conclusions

Considering all the limits of this anonymous questionnaire, in particular the possibility of misunderstanding the questions and, consequently, giving wrong answers, it allowed a glimpse of the influence of changes in daily life habits during and as a consequence of the period of the lockdown, on various areas of health. In particular, the population included in this study reported undergoing an increase in body weight, which is probably determined by the sedentary life and the minor sports activities due to the social restrictions. This could have resulted in a worsening of OSAS in 65% of patients with a previous diagnosis and with a significant number of people linking their TMJ/orofacial pain to their eating habits, albeit unspecified. This study confirmed how important it is to have a healthy diet in people with chronic conditions, such as OSAS and temporomandibular disorders. A Mediterranean diet can be a good solution, but also a lower intake of refined sugars can be helpful. A correlation between being infected with COVID-19 or the quarantine period with the worsening of painful symptoms of the temporomandibular joint was reported by the majority of the population studied. This can be caused by the high level of socio-economic pressure that people felt during that period, and it is known how this kind of disorder can be affected by the emotional status of patients. In some cases, due to long COVID-19 and its effects on the general health of people, there was a worsening of previous illnesses and disorders. According to this study, it is suggested to patients with these kinds of disorders, like OSAS and TMJ pain, to have a healthy lifestyle, reduce, as much as possible, stressful situations and improve their daily diet.

## Figures and Tables

**Figure 1 ijerph-19-07154-f001:**
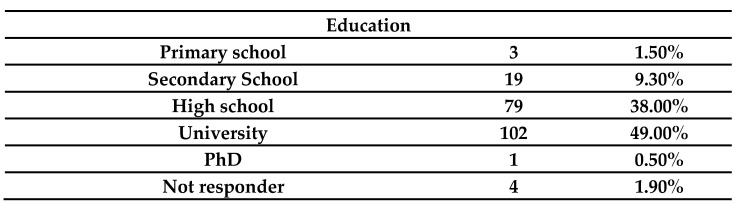
Education level of the interviewed population.

**Figure 2 ijerph-19-07154-f002:**
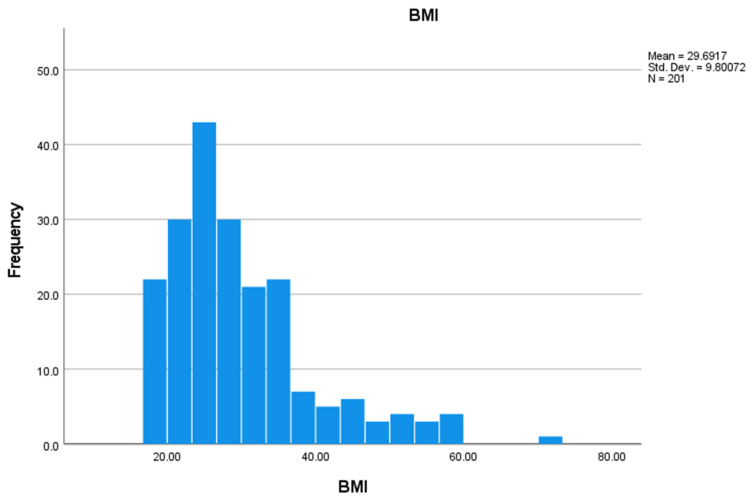
BMI of the analyzed sample.

**Table 1 ijerph-19-07154-t001:** Gender.

		Frequency	Percent	Valid Percent	Cumulative Percent
Valid	F	154	74.0	75.1	75.1
	M	51	24.5	24.9	100.0
	Total	205	98.6	100.0	
Missing		3	1.4		
Total		208	100.0		

**Table 2 ijerph-19-07154-t002:** Where do you live?

		Frequency	Percent	Valid Percent	Cumulative Percent
Valid	Africa	2	1.0	1.0	1.0
	Asia	8	3.8	3.9	4.9
	Australia	5	2.4	2.4	7.3
	Canada	4	1.9	1.9	9.2
	Europe	45	21.6	21.8	31.1
	India	1	0.5	0.5	31.6
	Italy	92	44.2	44.7	76.2
	Middle East	1	0.5	0.5	76.7
	North America	6	2.9	2.9	79.6
	South America	2	1.0	1.0	80.6
	USA	40	19.2	19.4	100.0
	Total	206	99.0	100.0	
Missing		2	1.0		
Total		208	100.0		

**Table 3 ijerph-19-07154-t003:** Did you get infected with COVID-19?

		Frequency	Percent	Valid Percent	Cumulative Percent
Valid	No	149	71.6	73.4	73.4
	Yes	54	26.0	26.6	100.0
	Total	203	97.6	100.0	
Missing		5	2.4		
Total		208	100.0		

**Table 4 ijerph-19-07154-t004:** Did you notice a change in your weight during the COVID-19 pandemic?

		Frequency	Percent	Valid Percent	Cumulative Percent
Valid	No	81	38.9	40.3	40.3
	Yes	66	31.7	32.8	73.1
	Yes. Gained weight	46	22.1	22.9	96.0
	Yes. Lost weight	8	3.8	4.0	100.0
	Total	201	96.6	100.0	
Missing		7	3.4		
Total		208	100.0		

**Table 5 ijerph-19-07154-t005:** Did you need the dentist during the pandemic?

		Frequency	Percent	Valid Percent	Cumulative Percent
Valid	No	74	35.6	36.6	36,6
	Yes	128	61.5	63.4	100,0
	Total	202	97.1	100.0	
Missing		6	2.9		
Total		208	100.0		

**Table 6 ijerph-19-07154-t006:** If Yes, did you go?

		Frequency	Percent	Valid Percent	Cumulative Percent
Valid	No	22	17.2	17.3	17.3
	Yes	105	82.0	82.7	100.0
	Total	127	99.2	100.0	
Missing		1	0.8		
Total		128	100.0		

**Table 7 ijerph-19-07154-t007:** Do you suffer from orofacial/TMJ pain?

		Frequency	Percent	Valid Percent	Cumulative Percent
	No	108	51.9	54.8	55.8
	Yes	87	41.8	44.2	100.0
	Total	197	94.7	100.0	
Missing		13	6.3		
Total		208	100.0		

**Table 8 ijerph-19-07154-t008:** Do you feel TMJ/orofacial pain is correlated with what you eat?

		Frequency	Percent	Valid Percent	Cumulative Percent
Valid	No	53	60.9	61.6	61.6
	Yes	33	37.9	38.4	100.0
	Total	86	98.9	100.0	
Missing		1	1.1		
Total		87	100.0		

**Table 9 ijerph-19-07154-t009:** Did your TMJ/orofacial pain increase during the pandemic?

		Frequency	Percent	Valid Percent	Cumulative Percent
Valid	No	41	47.1	48.2	48.2
	Yes	44	50.6	51.8	100.0
	Total	85	97.7	100.0	
Missing		2	2.3		
Total		87	100.0		

**Table 10 ijerph-19-07154-t010:** Did you notice an influence of your weight/diet on obstructive sleep apnea syndrome (OSAS)?

		Frequency	Percent	Valid Percent	Cumulative Percent
Valid	No	8	32.0	34.8	34.8
	Yes	15	60.0	65.2	100.0
	Total	23	92.0	100.0	
Missing		2	8.0		
Total		25	100.0		

## Data Availability

The data that support the findings of this study are available from the corresponding author (S.S.) upon reasonable request.

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
