# Peer review of "The Influence of SARS-CoV-2 Pandemic on TMJ Disorders, OSAS and BMI"

_ijerph, 2022, doi:10.3390/ijerph19127154_

Round 1
Reviewer 1 Report
Dear Authors,
your idea is interesting and well executed, but some improvements are needed. As you wrote, the study has many limitations. It would be adequate to better study the results with a stratification between 2 or more variables. (OSAS increase vs Age, OSAS and TMD, etc). In this way, the high risk of bias should be reduced. The possible causes of a worsening in OSAS or TMD are various: many statement in the paper are just hypotheses, even if well proved.
For these reasons, a deeper analysis of your results would increase the scientific significance of your paper.
Check the mismatch bitween italian and english in your tables.
Check to submit a clean manuscript without barred words “words”.
Author Response
the answer is in the file attached below.

Reviewer 2 Report
Rather, it should be TMD and not just TMJ, as stress and mental tension have a huge impact on the muscular cause of the disorder. The more so as the Authors themselves indicate the impact of the stress resulting from the pandemic on the quality of life, in addition to avoiding physical activity, and indicate an increase in the occurrence of orofacial pain.
Verse 35 is used twice as myocarditis.
In verse 45 the abbreviation is first used TMD, although previously it was used: TMJ disorders which is the narrower term.
In Table 8, the word "si" is used instead of "yes".
Do patients know the symptoms of OSAS - maybe the Epworth scale should have been used? Because there may be people who have started OSA symptoms. In the presented form, only people treated with OSAS could answer correctly whether they observed the intensity, especially since the youngest people participating in the survey were 11 years old. Similarly with questions about TMJ and orofacial pain - exactly what were these questions, because many people do not know where TMJ is, especially children. Please provide the questionnaire completed by patients as an attachment. That is, the authors only made the severity of TMJ and orofacial pain dependent on food and could only refer to people who had previously been treated for this disease and knew about the symptoms, as did people with OSAS (verse 126, 130).
In verse 109, the mean age should be added, including the SD. Please write down your inclusion and exclusion criteria: the questions only allowed to affect people with TMJ disorders / orofacial pain, OSAS, and only the weight can be checked for everyone. Please describe the statistical methods used, the level of significance. Only figures 1 and 2 are attached, the others are not. In table 7 there is a description: "sometimes mandibular joint" - what does that mean? Not to mention that there is a TMJ and not a mandibular joint. There is also a question about migraines after the 3rd dose of vaccination. Where did the question about primary headaches suddenly come from? Please describe it correctly.
What was meant by eating habits in the survey? Is the texture of the food - soft, hard, grains, or unhealthy food - high in sugar, gluten, etc.? Because in Results there is one interpretation and in Discussion there is another interpretation, similarly in Conclusion.
TMD is first used in verse 225! There is a haos because, in conclusion, the Authors give suggestions, based on their research, from which they do not arise. Please describe it clearly.
Please correct this and describe reliably what was tested, taking into account the differences between TMJ disorders and TMD, and stick to the terms used consistently. Describe statistical methods, correlations, levels of significance. The article requires re-review.
Author Response

(The authors gave the same response as above.)

Round 2
Reviewer 2 Report
I thank the Authors for the corrections. I just think that the tables in the first version were good, they only contained Italian words - please restore the tables only leave the current names and linguistic corrections. I have comments only on table 7 where the term "mandibular joint" is mentioned - please change it. In addition to these corrections, the article may be printed.
Author Response
Thank you for the new review. We are very pleased that our corrections were appreciated. We introduce the other corrections suggested in the last review.
We introduced again the tables of the previous version but revised. We corrected the term “ mandibular joint”, which was used by a specific patient.
